Linear Lepidopteran ambidensovirus 1 sequences drive random integration of a reporter gene in transfected Spodoptera frugiperda cells

Rizk Francine 1 2 3 francinerizk@gmail.com
Laverdure Sylvain 1 2 4
d’Alençon Emmanuelle 2
Bossin Hervé 2 5 6
http://orcid.org/0000-0002-2168-8268 Dupressoir Thierry 1 2 thierry.dupressoir@ephe.sorbonne.fr
1 EPHE, PSL Research University, UMR 1333 DGIMI, Université de Montpellier , Montpellier , France
2 UMR 1333 DGIMI INRA/UM, Université de Montpellier , Montpellier , France
3 Department of Life and Earth Sciences, Faculty of Sciences, Branch II, Innovative Therapeutic Laboratory, Lebanese University , Beirut , Lebanon
4 Laboratory of Human Retrovirology and Immunoinformatics (LHRI), Leidos Biomedical Research Clinical Services Program, National Cancer Institute , Frederick, MD , USA
5 Laboratoire d’Entomologie Médicale, Institut Louis Malardé , Papeete , French Polynesia
6 Aix Marseille Univ, IRD, AP-HM, SSA, VITROME, IHU-Méditerranée Infection , Marseille , France
Vassetzky Yegor
Electronic publication date: 2018 May 28
Publication date: 2018
Volume: 6
Electronic Location ID: e4860
Received 2018 Feb 12; Accepted 2018 May 4
Copyright: © 2018 Rizk et al.
Copyright year: 2018
Copyright holder: Rizk et al.
License: This is an open access article distributed under the terms of the Creative Commons Attribution License, which permits unrestricted use, distribution, reproduction and adaptation in any medium and for any purpose provided that it is properly attributed. For attribution, the original author(s), title, publication source (PeerJ) and either DOI or URL of the article must be cited.
License URL: https://creativecommons.org/licenses/by/4.0/

Keywords: Densovirus, Sf9, Expression, Integration, Linear

Funding: Agence Universitaire de la Francophonie (AUF) Francine Rizk was a recipient of an Agence Universitaire de la Francophonie (AUF) doctoral fellowship. The funders had no role in study design, data collection and analysis, decision to publish, or preparation of the manuscript.

==============================
Background

The Lepidopteran ambidensovirus 1 isolated from Junonia coenia (hereafter JcDV) is an invertebrate parvovirus considered as a viral transduction vector as well as a potential tool for the biological control of insect pests. Previous works showed that JcDV-based circular plasmids experimentally integrate into insect cells genomic DNA.

Methods

In order to approach the natural conditions of infection and possible integration, we generated linear JcDV-gfp based molecules which were transfected into non permissive Spodoptera frugiperda (Sf9) cultured cells. Cells were monitored for the expression of green fluorescent protein (GFP) and DNA was analyzed for integration of transduced viral sequences. Non-structural protein modulation of the VP-gene cassette promoter activity was additionally assayed.

Results

We show that linear JcDV-derived molecules are capable of long term genomic integration and sustained transgene expression in Sf9 cells. As expected, only the deletion of both inverted terminal repeats (ITR) or the polyadenylation signals of NS and VP genes dramatically impairs the global transduction/expression efficiency. However, all the integrated viral sequences we characterized appear “scrambled” whatever the viral content of the transfected vector. Despite a strong GFP expression, we were unable to recover any full sequence of the original constructs and found rearranged viral and non-viral sequences as well. Cellular flanking sequences were identified as non-coding ones. On the other hand, the kinetics of GFP expression over time led us to investigate the apparent down-regulation by non-structural proteins of the VP-gene cassette promoter.

Conclusion

Altogether, our results show that JcDV-derived sequences included in linear DNA molecules are able to drive efficiently the integration and expression of a foreign gene into the genome of insect cells, whatever their composition, provided that at least one ITR is present. However, the transfected sequences were extensively rearranged with cellular DNA during or after random integration in the host cell genome. Lastly, the non-structural proteins seem to participate in the regulation of p9 promoter activity rather than to the integration of viral sequences.

Introduction

Ambidensoviruses belong to the Parvoviridae family, a group of small, non-enveloped viruses with icosahedral symmetry and a linear genome of 4–6 kb single-stranded (ss) DNA. Parvoviridae are divided into two subfamilies, depending upon whether they infect vertebrates (Parvovirinae), or invertebrates (Densovirinae, DNV). The latter currently include 15 virus species, divided into five genera, Ambidensovirus, Brevidensovirus, Hepandensoviruses, Iteradensovirus and Penstyldensovirus (Cotmore et al., 2014) even if that classification will probably be modified after the finding of echinoderm-infecting “densoviruses” (Hewson et al., 2014). DNVs, like other members of the Parvovirus genus, are autonomously replicating viruses and differ in this respect from members of the genus Dependoparvovirus which need helper functions for replication. The DNV isolated from the lepidoptera Junonia coenia belongs to the species Lepidopteran ambidensovirus and will be named JcDV hereafter. JcDV (Junonia coenia densovirus, isolate Oxford, GenBank: KC883978.1) has a 6,032 nucleotides genome with identical 547 nucleotides inverted terminal repeats (ITR). It exhibits an ambisense genomic organization, the non-structural (NS) and structural (VP) genes being positioned in the 5′-half of each complementary strand after conversion of the ss infectious genome in a double-stranded replicating genome. Each strand is encapsidated separately and converted into double-stranded DNA after infection (Dumas et al., 1992). The JcDV genome ambisense organization displays a major open reading frame (ORF1) encoding four capsid proteins VP1 to VP4, the expression of which is under the control of the VP-gene cassette promoter, hereafter designated p9, on one strand (Wang et al., 2014), and three ORFs coding for replication proteins NS-1, NS-2 and NS-3 are located on the complementary strand and controlled by the NS-gene cassette promoter, traditionally designated p93 promoter (Wang et al., 2014). Those are translated from a 2.4 kb genomic mRNA (NS-3) and a sub-genomic 1.7 kb mRNA (NS-1 and NS-2) respectively (Wang et al., 2014).

The potential of Densovirinae as biocontrol agents against insect pests has been reported (Belloncik, 1990; Bergoin & Tijssen, 1998; Fediere, 2000; Tal & Attathom, 1993). Ambidensoviruses, as well as other members of the Densovirinae subfamily, have also been used as gene transfer vehicles to investigate viral pathogenesis and genetic manipulation of insects after integration of plasmidic ambidensovirus vectors either in vitro and in vivo (Afanasiev & Carlson, 2000; Bossin et al., 2003, 2007; Gu et al., 2011; Royer et al., 2001). On the other hand, dependoparvoviruses as well as Rodent protoparvovirus 1 DNA molecules were shown to integrate in a foreign DNA after recognition, by the replication proteins, of precise binding and cutting sites shared between target genomic sequence and viral hairpins (Corsini, Tal & Winocour, 1997; Hendrie, Hirata & Russell, 2003; Janovitz et al., 2013). Last, recent data show that a significant amount of Parvoviridae DNA remnants sign past infections, some of which gave rise to endogenization and expression of viral proteins (Liu et al., 2011). JcDV ITRs as well as non-structural protein 1 (NS1) activities resemble those of “integrative” Parvovirinae (Ding et al., 2002) and could explain the above mentioned integration events of plasmid DNA sequences in vitro and in vivo (Bossin et al., 2003, 2007; Royer et al., 2001).

In order to evaluate the capacity of linear JcDV sequences, more relevant to infectious viral genome, to integrate within the host DNA, different non-infectious linear constructs encompassing JcDV sequences were transfected into the non-permissive Sf9 cell line. These constructs express the green fluorescent protein (GFP) gene under the control of the JcDV p9 promoter, which allowed us to follow the integration and expression of linear JcDV genes. In addition, our observations led us to investigate a possible role of JcDV NS proteins on the activity of the p9 promoter in vitro.

Altogether our results show that defined JcDV-derived sequences brought by linear DNA molecules are able to drive efficiently the integration and expression of a foreign gene into the genome of insect cells. We confirm that, as already described with circular plasmid vectors, some viral sequences—i.e., ITRs and regulatory regions around the polyA signals—are indispensable for a correct expression of the transgene. Some viral integrated sequences as well as their flanking chromosomal regions were analyzed and were interpreted as a randomized integration within repeated or transposable elements (TE) and/or intergenic regions. Viral sequences show dramatic rearrangements in all cases and no complete viral genome was recovered from the integrated forms which were characterized. However, our results do not fully sustain the previous assessment that NS proteins expression favors integration at a low copy number in the host genome. In addition, a stable NS gene expression seems to be accompanied by a lower expression of the reporter gene. This result led us to evaluate in vitro a possible trans-regulation of the p9 promoter by the NS proteins, which does not seem to be the case. Our results led us to conclude that a recombinant JcDV virus encompassing only a transgene and an ITR could be sufficient as an integrative expression vector in insect cells.

Materials and Methods

Plasmids and derived linear molecules

Linear molecules were derived from already described JcDV-based plasmids, pJGFPH (pFull) encompassing an infectious sequence of the JcDV genome and pJGFPΔNSH (pVP) with deleted NS genes. For both plasmids the GFP coding sequence and simian virus 40 polyadenylation signal are inserted so as to place the gfp in frame, 25 codons downstream of the ATG initiation codon for VP4 polypeptide, under the transcriptional control of the P9 viral promoter (Bossin et al., 2003). pHp9GFPNSH (pNS) vector displays the genuine gfp start codon 23 nucleotides downstream p9 TATA box.

For the measurement of the p9 promoter activity, the reporter plasmid (pHp9LUC-DNS) was obtained as follows: pFull was digested with NdeI then re-ligated to eliminate the p93 promoter and most of the NS proteins coding sequence. The gfp was then replaced with firefly luciferase gene (Luc), amplified from the pGL3 basic vector (Promega Madison, WI, USA) with the primers 5′-TGTTGGTAAAGCCACCATGGAAG-3′ (sense) and 5′-CTCGAAGCGGGCGGCCGCCCCGACTCTAG-3′ (antisense) using a high fidelity polymerase (Platinum Taq DNA Polymerase; Invitrogen, Courtaboeuf, France). A transfection control expressing the Renilla reniformis luciferase (Nalcacioglu et al., 2003) under the control of the IE1 promoter of Autographa californica Nucleopolyhedrovirus (AcNPV) (kind gift of Just Vlak, Wageningen University, The Netherlands) was used as a “relative transfection control” of transduction efficiencies. Co-transfections were performed with JetPEI (Polyplus-Transfection, Illkirch, France) and luciferase activity was revealed using the Dual Luciferase Promoter Assay System (Promega, Madison, WI, USA) and quantified on a Berthold Centro LB 960 luminometer; Berthold Technologies, Thoiry, France.

A3GFP plasmid was derived from A3-LacZ construct (Mange et al., 1997) after replacement of the lacZ gene sequence with the corresponding gfp one (thanks to F.X. Jousset, INRA).

Linear molecules were obtained after proper restriction of 10 μg of JcDV-based pFull, pNS and pVP vectors. DNA was cut as indicated in Fig. 1 with restriction enzymes ClaI, PvuII, XcmI, AflII, MspA1L and PshAI according to the New England Biolabs protocols. ClaI and PvuII cut 163 nucleotides upstream JcDV’s right ITR and 333 nucleotides upstream left ITR, respectively in the pBR322 plasmid backbone whereas XcmI cuts eight nucleotides upstream the p93 TATA box. AflII restriction site deleted the polyadenylation signals (poly-A) which belong to both ORF1 and ORF2/ORF3 in the ambisense organization of JcDV and MspA1L and PshAI cut within right and left ITRs at positions 526/5506 and 88/5944, respectively according to the nts numbers in the virus sequence. Restricted DNA was separated on a 0.8% agarose gel and expected bands were cut out from the gel (NucleoSpin® Extract II kit; Macherey-Nagel, Düren, Germany). Restricted DNA size and purity were assessed following conventional protocols.

Figure 1 Schematic organization of JcDV-based plasmids used to generate linear sequences for transfection experiments.

pBR322 backbone is figured as dotted gray line. JcDV structural proteins (VP) coding sequences are represented by a solid black line; non-structural proteins (NS) genes by a solid gray line. Both share a polyadenylation signal shown as an open ellipse. Open boxes figure the p9 and p93 ITRs, hatched boxes underline the location of p9 and p93 promoters, respectively. GFP coding sequence is figured with a gray box and its 3′ SV40-derived polyadenylation signal is shown as a hatched line. Arrows numbered according to Table 1 figure the primers used for PCR-based experiments. Primers used for walk-PCR are represented above; primers 7–13 represented only relatively to p9 can also hybridize to p93 DNA sequences when present. Gray-filled arrowheads figure restriction sites used to generate linear molecules from the plasmid constructs. Open arrowheads figure restriction sites used for walk-PCR experiments. Subscript numbers indicate iterated restriction sites. By convention, nucleotide numbers of each linear molecule are accorded to the 5′ C generated after ClaI restriction (AT/CGAT). Restriction enzymes are: A: AflII, C: ClaI, D: DraI, H: HpaI, M: MspA1L, S: SspI, P: PshAI, Pv: PvuII, X: XcmI, respectively. Linear molecules were obtained after restriction of three different JcDV-based vectors. Their length is indicated under the name of each plasmid: (A) pFull encompassing a full-length sequence of JcDV DNA and the GFP marker gene, cloned into pBR322 plasmid. This schematic representation displays all the symbols described above; some of them only are reported in B and C. (B) pVP in comparison to pFull, a frameshift deletion affects the NS region (C) pNS in comparison to pFull, lacks VP genes. The expression of GFP is directly under the control of the p9 promoter. Primers giving rise to specific products after PCR are shown (See Table 1).

Cell culture and transfection system

Sf9 cells (ATCC CRL 1711) derived from S. frugiperda ovaries (Vaughn et al., 1977) and IPLB-Ld652 cells derived from Lymantria dispar ovaries (Goodwin, Tompkins & McCawley, 1978) were maintained at 28 °C in 10% heat-inactivated fetal calf serum (FCS) supplemented TC100 medium.

A total of 1 μg of DNA was used for the transfection of 2.104 Sf9 cells with Jet PEI (Polyplus-Transfection, Illkirch, France) according to the manufacturer’s protocol.

The number of GFP expressing cells, among 10,000 sorted live cells, was measured via fluorescent activated cell sorter (BD FACSCalibur; BD Niosciences, Le Pont de Claix, France) at 488 nm excitation and 530 nm emission. Microscopy was performed with a Nikon MICROPHOT-FXA microscope /ACT-2U software (Nikon, Tokyo, Japan).

Fluorescent colonies were picked out then amplified to confluence in Sf9-conditioned TC100 medium supplemented with 20% heat-inactivated FCS. Each amplified clonal cell population was diluted and subcloned three times before populations stably expressing GFP were obtained (approximately two clones per initial transfection).

DNA analysis

Low molecular weight DNA was extracted according to Ziegler et al. (2004). For genomic DNA analysis, 30 μg of DNA were extracted using standard protocols and restricted with 150 U of the appropriate restriction enzymes. All the restriction enzymes were from New England Biolabs. Restricted DNA was separated on a 0.8% agarose gel and bands were cut out from the gel (NucleoSpin® Extract II kit; Macherey-Nagel, Düren, Germany). Size and purity of each DNA fragment were assessed electrophoretically. Digested DNA were prepared for Southern blotting analysis according to conventional methods (Sambrook, Fritsch & Maniatis, 1989). Blots were then probed with a full length (pFull restricted with PshAI) probe, either Dig-labelled for episomal DNA analysis, or ΔdCT32P-labelled for Southern blot analysis. Hybridizations were performed according to the respective manufacturer’s protocols. Radioactive membrane was exposed on a storage phosphor screen and visualized using a Phosphor Imager STORM 840 after two or five days.

Statistical analysis

The percentage of GFP expressing cells from five different measurements at three, six, 10, 18 and 30 days post-transfection (p.t.) was measured. The evolution of the GFP positive cells percentage along with time was represented with a linear correlation curve (R2 > 0.5) and graphically compared. In addition, a Wilcoxon rank sum test with continuity correction (R suite; R Core Team, 2014) to the raw fluorescent cells number obtained after transfection of plasmid vs. linear JcDV-based vectors at four time points (6, 10, 18 and 30 d., p.t.).

Significance of differences in luciferase expression was assessed by a Student test (R suite; R Core Team, 2014).

PCR and RT-PCR

A total of 100 ng of DNA from each analyzed cell clone was used as a template for direct PCR (see Table 1; Fig. 1 for primers names, sequences and positions). The VP-GFP fusion was assessed with primers located in the VP1 (1) and in the C-terminus VP sequence (2) downstream the gfp gene. The right ITR (p9 ITR)-GFP region was examined in clones transfected with pNS-based molecules, using primers located in the p9 ITR (1′) and the gfp gene (2′). The NS-GFP region was amplified with primers based in the gfp (3) and NS-3 genes (4). The long p9 ITR-NS region was assessed with primers located in the p9 ITR (1′) and the NS-3 (4), using a long range/high fidelity amplification DNA polymerase (Herculase; Stratagene Agilent Technologies, Les Ulis, France).

Table 1 Primers used in PCR, RT-PCR, reverse PCR and walk PCR analysis.

Analysis	Primer	Sequence	
PCR (VP-GFP)	1
2	5′-TAGTCAGTATGTCTTTCTACACGGC-3′
5′-AACGGTGGTTTAATTAAACCC-3′	
PCR (ITR-GFP)	1′
2′	5′-GTGACCTCGTTTGACCGGC-3′
5′-GCTGAACTTGTGGCCGTTTAC-3′	
PCR (NS-GFP)	3
4	5′-GCATGGACGAGCTGTACAAG-3′
5′-GTTTCTTTGTGTTCGTCGTTTATTTG-3′	
RT-PCR (NS)	5
6	5′-CGTCCAAACATTGATCACGGAGCTG-3′
5′-GTAGTGTTGTGCAAAAGTGGTTCCAGA-3′	
Reverse PCR	11	5′-GGTCAAACGAGGTCACAATAACAAGA-3′	
WalkPCR(Adaptor sequences)	ADPR1
ADPR2	5′-CTAATACGACTCACTATAGGGCTCGAGCGGCCGCCCGGGGAGGT-3′
5′ P-ACCTCCCC-3′NH2	
(Adaptor primers)	AP1
AP2	5′-GGATCCTAATACGACTCACTATAGGGC-3′
5′-CTATAGGGCTCGAGCGGC-3′	
(P9-ITR region)	13
12	5′-CTGTTTTGCACACGGCCCAG-3′
5′-CCAGCCTCGACGCGAGTTTG-3′	
(ITR region)	10
9
8
7	5′-GCCGGTCAAACGAGGTCAC-3′
5′-GCTATCTCGCTCTAACAGTTGC-3′
5′-CTCGCACACTATTGCTGTCCTTC-3′
5′-CAGCTCCAAGGTCTTCGGATC-3′	
(VP1 region)	15
14
19	5′-CCAAGTTCAATATCTTCAGTAGCAGTAC-3′
5′-GATGTATTAACCCGGCCGTGTA-3′
5′-CCTATGATTCCCACTGCTACTAGT-3′	
(GFP region)(NS3 region)(UTR region)	16
17
18	5′-GCAGCTTATAATGGTTACAAATAAAGC-3′
5′-CAAATAAACGACGAACACAAAGAAAC-3′
5′-TCACTGAGATGTTCACTCGAC-3′	

Total RNA was extracted from cells (NucleoSpin RNA II kit; Macherey-Nagel, Düren, Germany). The retrotranscription of mRNA was performed using the ThermoScript RT-PCR System kit (Invitrogen, Courtaboeuf, France). Amplification of the NS transcripts from the cDNA templates was performed using primers located in the NS-1 (5) and the untranslated region 5′ to NS-3 (6).

PCR were performed on a GeneAmp PCR system 2700; Applied Biosystems, Illkirch, France, using conditions adapted to the melting temperatures of the primers and expected sizes elongations.

Reverse PCR and walk PCR

Reverse PCR was performed according to Ochman, Ayala & Hartl (1993). Briefly, 10 μg of cellular DNA was digested with HpaI O/N at 37 °C and ligated O/N at 4 °C (T4 DNA ligase kit; Promega, Madison, WI, USA). PCR was performed on the circularized DNA with primers located in the VP1 (14)/gfp (3), for lpFull and lpVP clones and in right ITR (11)/gfp (3) for lpNS clones lacking VP sequences. PCR products were run on a 0.8% agarose gel, purified (NucleoSpin Extract II kit, Macherey-Nagel, Düren, Germany) and sent for sequencing to the service provider (Genome express; see Table 1; Fig. 1 for primers names, sequences and positions).

For walk PCR assays, genomic DNA from GFP+ clones was digested with either SspI, HpaI or DraI (several restriction sites in the viral sequence) or AvaI or EcoRV (non-cutter enzymes). A specific double-stranded adaptor with cohesive ends ADPR1/ADPR2 was allowed to hybridize before ligation O/N at 37 °C (T4 DNA ligase kit; Promega, Madison, WI, USA) at each extremity of the previously restricted DNA fragments. Then, nested PCRs were performed according to Sallaud et al. (2003). A set of nested primers located in the ITR sequence was designed (7, 8, 9, 10, 12 and 13) and two nested primers were selected in the adaptor sequence (AP1, AP2). Each adaptor-ligated fragment was submitted to a series of primary and nested amplifications. EcoRV or AvaI restricted genomic fragments were submitted to ligation and amplification as described above. For walk PCR analysis, two overlapping primers located in the VP1 sequence (14 and 15) or in the gfp (three and 16) or in the NS-3/UTR (17, 18), were used to combine with AP1 and AP2 primers. PCR products were purified and sent for sequencing as above.

Data mining

Raw sequences from the amplified fragments were compared to each other, assembled using Cap 3 (http://doua.prabi.fr/software/cap3) when necessary and submitted for comparison to public databanks NCBI (http://www.ncbi.nlm.nih.gov/BLAST/) (Altschul et al., 1997),

SPODOBASE (http://bioweb.ensam.inra.fr/spodobase/) and

Lepido DB (http://bipaa.genouest.org/blast/sfru/#)

and aligned against the JcDV, isolate Oxford (GenBank: KC883978.1) using the Megablast conditions for viral sequences and the blastn conditions for the genomic ones. Only the results obtained with an expected value > 10e5 are taken into consideration.

Results and Discussion

JcDV-based linear vectors drive stable expression of GFP in Sf9 cell lines

pFull (pJGFPH), pVP (pJGFPΔNSH) and pNS (pHp9GFPNSH) circular plasmids were used as positive controls for transfection and GFP expression in Sf9 cells. The same plasmids were properly restricted to generate linear molecules encompassing different regions of the viral sequences and the gfp as a marker gene (Fig. 1).

After transfection, expression and accumulation of GFP were monitored under microscopic examination up to six days p.t. then the percentage of GFP positive cells (GFP+) was estimated by visual counting of three different fields (∼100 cells).

An unrelated control, devoid of JcDV sequences, consisted of plasmid A3GFP which contains the gfp under the control of late actin A3 ubiquitous promoter of Bombyx mori. Transfection of both circular and linearized A3GFP molecules gave rise to a small, rapidly decreasing number of fluorescent cells which was undetectable one day p.t.. Although GFP+ cells appeared 48 h p.t. using circular plasmids, their relative number decreased along with time in agreement with previous observations (Bossin et al., 2003). Surprisingly, for most linearized molecules, appearance of GFP+ cells showed a delay before reaching at 6 d. p.t. percentages equivalent to those obtained with circular molecules.

Complete deletion of both right- and left-ITRs (p9 and p93 ITR, respectively, MspA1L) gave rise to rare GFP positive cells which persisted after one month in culture. The resulting expression of GFP can be related to the integration of transfected molecules in a transcribed genomic region.

When pFull, pVP and pNS were restricted with ClaI + AflII the GFP+ cell number decreased dramatically. Since the ClaI cutting site, located in the pBR322 backbone, is used to generate every linear molecule, this effect is probably due to the AflII cutting site which was brought along with the gfp gene during the construction of plasmids. Restriction with AflII deleted the polyadenylation signals (poly-A) which belong to both ORF1 and ORF2/ORF3 in the ambisense organization of JcDV (Dumas et al., 1992; Shirk et al., 2007) and are present in all three constructs. This result is in agreement with the pivotal role of Polycomb Response Elements in this region, as described by Shirk et al. (2007).

We subsequently chose three linear molecules allowing high transformation efficiency, for further comparison with the corresponding circular plasmids.

pFull as well as pVP, were digested with ClaI + PvuII which deleted the pBR322 plasmid backbone. lpFull expresses either NS proteins and VP-GFP proteins; lpVP exhibits a deletion within NS sequences.

pNS was linearized with ClaI + XcmI to obtain the lpNS encompassing the p9 ITR, gfp and the NS sequences, but not the VP ones. Noticeably, deletion of the p93 ITR (ClaI + XcmI restrictions) seemed not to affect the percentage of cells in which GFP accumulates. XcmI cuts a few nucleotides upstream the p93 TATA box and still allows expression of ORF2 ORF3 and ORF4 (see below).

In each pair of linear molecules and circular corresponding ones, the gfp marker gene is expressed, either as a VP-GFP fusion protein (pFull, lpFull; pVP, lpVP) or under the direct control of the p9 capsid protein promoter (pNS, lpNS) (Fig. 1).

Sf9 cells were transfected in parallel with pFull/lpFull, pVP/lpVP and pNS/lpNS. The percentage of GFP-fluorescent cells was followed as a function of time p.t.. The results, shown in Fig. 2, of three to six independent experiments, allowed the establishment of linear regression profiles of the GFP+ cells percentages, at 3, 6, 10, 18 and 30 d. p.t.. All linear constructs gave significant GFP expression starting six days p.t.. Transfection of lpFull gave rise to an increasing number of GFP+ cells over the next 20 days to finally reach, at 30 days p.t., a 10% plateau. lpVP transfection gave rise to an increasing number of GFP+ cells reaching 11% at 30 days p.t.. The lpNS construct allows the obtention of 4% GFP+ cells at 30 days p.t.. As expected, VP-GFP expression was visible as a bright punctate staining in the nucleus of lpFull and lpVP transfected cells (Fig. 3A) although pNS circular and linear molecules gave rise to a cell widespread GFP expression (Fig. 3B). A scheme of individual cell populations obtained after three cloning rounds is shown in Fig. 3C.

Figure 2 Relative enrichment of GFP+ cells along with growth of transfected Sf9 cells.

Following transfection of Sf9 cells using Jet PEI and indicated DNA molecules, GFP expression was measured in cell populations three days, 6 d., 10 d., 18 d. and 30 d. post-transfection (p.t.) by flow cytometry. The percentage of GFP+ cells was measured in three to five independent experiments and the results are reported on each coordinates. Linear regression curves can be drawn which show the evolution of GFP+ cells percentages along time (R2 > 0.5). a, non significative; b, p < 0.05; c, p < 0.01 (Wilcoxon rank sum test with continuity correction, R suite; R Core Team, 2014).

Figure 3 Cellular clones showing persistent expression of GFP after transfection with JcDV-based linear molecules.

(A) Typical clump of cells transfected with lpFull or lpVP and showing mainly nuclear localization of the GFP fluorescence. (B) Widespread cytoplasmic fluorescence in lpNS transfected cells. (C) Isolation scheme of clonal cell populations from clumps exemplified in A and B. Numbers between brackets are the percentage of fluorescent cells in each clone as measured by cytometry. Several stable fluorescent cell clones were recovered and designated lpX-n1Yn2, “l” is for linearized, “pX” for the plasmid origin of the molecule and the combination “n1Yn2” depicts the steps of clonal population isolation.

Along with cellular divisions, the percentage of GFP-positive cells increased in cells transfected with linear molecules to reach, 30 days p.t., an amount three to four times higher than in cells transfected with circular constructs. It could be hypothesized that early integration events are “stabilized” along with time, meaning that the gfp expression becomes more efficient when the p9 promoter is surrounded with a favorable molecular environment, for instance after integration in the chromosomal DNA of the host cell. It is also possible that the p9 promoter activity is regulated along time after integration. The latter possibility is further explored below.

Stable GFP-expressing cell clones from both circular and linear molecules transfection were obtained, but only lpFull, lpVP and lpNS transfected cells were used to generate sub-clones derived from a single transfected cell. It is to be noted that along with the cloning processes, GFP expression decline was observed in 10% of clones due to gfp gene loss as revealed by PCR (not shown). From the rapidly growing cloned populations which were retained for further studies, all exhibited a sustained expression of GFP and were evaluated circa 90% GFP+ 12 months after cloning (FACS measurement, not shown).

JcDV sequences drive integration of vectorized DNA sequences in Sf9 transfected cells

To investigate the presence of episomal forms of transfected linear JcDV DNA-based molecules, low-molecular weight DNA from several stable cell clones (lpFull-3f2, lpFull-4a1, lpNS-6f1, lpNS-6f2 lpVP-9f3 and lpVP-9f4) was analyzed by Southern blotting (Fig. 4A). No signal was detected when probing with a DIG-labeled linearized JcDV probe suggesting that episomal forms were not detected in the tested clones nor remnant unintegrated linear molecules persisted in the clones.

Figure 4 Integration patterns are dependent on transfected viral sequences.

(A) Low molecular weight (LMW) DNA was extracted from two different stable subclones per linear construct and hybridized with a DIG-labelled probe. pB, pVP and pFull represent pBRJH (Dumas et al., 1992), pVP and pFull native plasmids, respectively. Sf9h, pBh, pVPh and pHh indicate LMW DNA extracted from mock-transfected, pB, pVP and pFull transfected Sf9 cells, respectively. (B) Genomic DNA was isolated from stable fluorescent cell sub-clones, restricted with either AvaI (A) or AvaI + HpaI (A + H) and separated electrophoretically. Hybridization was performed using a radioactively labeled probe. pFull plasmid and the subsequent linearized lpFull are used as positive control. Sf9 DNA was used as a negative control.

To check for integration events, DNA extracted from sub-cloned cell populations, were subjected to Southern Blot analysis (Fig. 4B). Genomic DNA was obtained, digested with either AvaI, or AvaI and HpaI, the latter cutting twice within lpFull and lpVP transgene sequences and once in lpNS molecules (AvaI does not cut within the transgene sequences, see Fig. 1). Restricted DNA was hybridized with a full-length 32P-labelled JcDV probe.

lpFull clones (lpFull-3f2, lpFull-4a1) DNA, digested with AvaI and hybridized to the probe, showed a single band at 12 kbp and 11 kbp respectively, larger than the 7.5 kbp lpFull transfected construct. Integration events in clones lpFull-3f2 and lpFull-4a1 thus occurred at a single but different site in each clone. After double-restriction with AvaI and HpaI, one smaller band is generated from the DNA of either lpFull-3f2 (5.5 kbp) or lpFull-4a1 clone (7 kbp). In the latter restriction, each of both obtained bands is larger than the 3 kbp restricted lpFull transgene sequence suggesting different integration sites in the two clones. These data suggest that lpFull sequence integration into lepidopteran cell line Sf9 was systematically associated with rearrangements.

DNA from the lpVP-9f3 clone (lacking NS genes), restricted with AvaI and Southern-blotted, showed two large bands at 13 kbp and 15 kbp larger than the 6.7 kbp lpVP transfected DNA suggesting more than one integration site. The analysis of lpVP-9f3 clone after restriction with AvaI and HpaI, shows at least six bands at 2.6, 2.8, 5.0, 6.0, 9.0 and 11 kbp smaller or larger than the 3 kbp restricted lpVP transgene sequences. Once again, this indicates that the transfected linear molecules underwent drastic modifications during or after integration in the recipient cell genome. Altogether, the presence of several bands suggests that viral sequences are integrated at different sites into Sf9 genome.

Consistent with previous observations after transfection of circular plasmid molecules (Bossin et al., 2003), lpFull, which is supposed to retain the capacity of coding NS proteins, gave rise to an apparent small number of integrated molecules. This could be interpreted as a single integration event whereas the analysis of clones obtained after lpVP transfection (modified NS genes) evoked possible multiple integration sites. We could not confirm concatemeric integration (Bossin et al., 2003).

Hybridization of the lpNS-6f2 clone DNA, after digestion with AvaI, revealed four bands estimated larger than 15 kbp and, at least, two additional bands between 10 and 15 kbp, which are much larger than the 4.1 kbp of the linear lpNS transfected sequence. It is to be noted that the hybridization signal is much more intense than that observed with lpFull- and lpVP-clones, for the same amount of total DNA. This could suggest that more integration events occurred when using lpNS molecules for transfection.

After digestion with AvaI and HpaI, lpNS-6f2 hybridization profile shows a 2.6 kbp band and 10 additional bands ranging from 2.2 kbp to 12 kbp. The revealed bands are all larger than the 2 kbp and 3 kbp lpNS restricted transfected sequences. Linear viral sequences also could probably be heavily rearranged after integration in numerous different sites.

To further investigate the organization of integrated viral sequences into Sf9 genome, PCR and RT-PCR amplifications were performed on the DNA extracted from GFP+ cell clones.

Integrated JcDV-derived sequences are rearranged

Primers (Table 1) were designed to detect two major viral target regions: the VP-gfp and the gfp-NS sequences (Fig. 1). Additional primers (1′, 4; cf. Fig. 1) were used to amplify a larger DNA fragment between the p9 ITR and the NS region.

In order to assess the presence of NS and VP sequences in transfected clones, DNA from a selection of representative clones for each transfection was subjected to PCR analysis. The 3 kbp VP-gfp (1, 2) region was successfully amplified from the cellular DNA of all lpFull (lpFull-3f2, lpFull-4a1) and lpVP (lpVP-9f3, lpVP-9f4) subclones, thus demonstrating that at least one copy of this region remains unaffected after integration (Fig. 5A). However, several unexpected amplified bands indicate that rearranged copies of this large molecular region are also present within the integrated JcDV-derived sequences in lpVP clones.

Figure 5 NS genes sequences are highly rearranged in stable fluorescent cell clones.

(A) PCR amplification of the VP-GFP region with primers one and two (see Table 1; Fig. 1) from lpFull and lpVP clones. (B) Amplification of the large p9 ITR-NS region using primers pair 1′, 4. (C) Amplification of the NS-GFP region using primers pair 3, 4. (D) Amplification of ORF2, ORF3 and ORF4 RNA transcripts by RT-PCR using primers in the untranslated region (UTR) upstream the NS-3 ATG codon and the NS-1 region (see Table 1). Molecular weight marker (Mk) band sizes are shown on each gel. pFull was used as positive control. Sf9 DNA was used as a negative control.

Using primers 1′ and 4, the expected p9 ITR-NS 3.4 kbp fragment is amplified from clone lpNS-6f2 DNA only (Fig. 5B). lpFull-3f2 and lpVP-9f3 DNA did not support the amplification of the expected 6 kbp and 5.1 kbp fragments respectively (Fig. 5B). Since lpVP-9f3 and lpFull-3f2 DNA gave rise to the amplification of 3.2 kbp and 3 kbp fragments respectively, the target DNA is likely to be affected by large deletions.

This was confirmed since NS-gfp region amplification (primers 3/4, Fig. 1) generated smaller bands than expected from the cellular DNA of lpFull-3f2 and lpFull-4a1 clones (expected band: 3.7 kbp, amplified: 0.4 kbp) and in lpVP-9f3 and lpVP-9f4 clones (expected band: 3 kbp, amplified: 1 kbp). Furthermore, in lpNS-6f1 and lpNS-6f2 clones, the expected 2.6 kbp band was properly amplified as well as an additional band at 2 kbp (Fig. 5C). These data show that at least one complete NS-gfp sequence is present within the genome of the lpNS transfected clones, whereas lpFull and lpVP DNAs do not support PCR amplification of this region.

Altogether, these results suggest that all lpFull, lpNS and lpVP clones, which were recovered and amplified, contain at least one copy each of a VP- or p9-gfp sequence stably integrated within the Sf9 genome whereas the repeated failure to amplify NS gene sequences from lpFull and lpVP clones indicate that major modifications, probably deletions, has occurred in this region along with the integration and/or cell cloning process. Only in lpNS clones one copy of the p93 ITR-NS region remained intact after integration.

In order to verify that the full NS copy could be functionally expressed, total mRNA was extracted from all stable clones and tested by RT-PCR for VP-GFP fusion transcripts as well as for NS transcription (Fig. 5D). Even if the RT kit involves a DNAse step, the RNA samples were verified DNA free according to direct PCR (not shown).

As expected, VP-GFP fusion mRNA, was successfully retrotranscribed then PCR-amplified in all lpFull, lpVP and lpNS clones (data not shown).

Only the mRNA extracted from lpNS-6f2 cells and subjected to RT-PCR amplification of the UTR-NS-1 region, gave rise to amplification of both the expected 400 bp (spliced form) (Abd-Alla et al., 2004) and 1,080 bp (unspliced form) bands, indicating that the 2.4 kb and 1.7 kb NS transcripts can be expressed in lpNS clones. An additional 800 bp band suggests that an unidentified mRNA transcript is also expressed from one of the numerous integrated JcDV-based sequences after transfection of Sf9 cells with lpNS molecules.

Attempts to amplify the 400 bp and/or the 1,080 bp band failed with cDNA obtained from lpFull and lpVP clones (Fig. 5D).

Rearranged viral regions are interspersed with non-viral sequences in transfected Sf9 cells

In order to, identify and characterize more precisely the rearrangement pattern as well as some of the junctions between integrated virus-based molecules and cellular Sf9 DNA, stable fluorescent clones resulting from lpFull, lpNS and lpVP transfections were analyzed using reverse PCR (Ochman, Ayala & Hartl, 1993) and walk-PCR (Sallaud et al., 2003). Each DNA sequence obtained after reverse- or walk-PCR was verified, by direct amplification of the cellular DNA extracted from JcDV-transfected clones, to generate a properly sized band (not shown). The raw sequences are presented in a .fasta format in Supplemental Information and Figs. 6A–6C summarize the global repartition of “scrambled” viral integrated sequences according to their respective positions in transfected constructs.

Figure 6 Schematic representation of viral and genomic DNA sequences identified in our experiments.

Raw sequences obtained after numerous amplifications of viral-genomic junctions (Supplemental Information) where aligned against (i) the sequence of the transfected linear plasmid using the discontiguous megablast conditions, (ii) the nr/nt nucleotide collection specified to Spodoptera frugiperda (taxid 7108), using the blastn conditions. Only the cleaned sequences are represented and motifs identify contiguous sequences from the same amplified sequence, named on the right. (A) BLAST-based alignment against lpFull of raw sequences obtained after amplifications of viral junctions from lpFull-transfected cells. The open diamond represents the location of ∼20 nts from the Bac 68E14 from S. frugiperda [id:681381.1] as a significative example of the rearrangement. (B) BLAST-based alignment against lpVP of raw sequences obtained after amplifications of viral junctions from lpVP-transfected cells. The triangle represents 70 nts of BAC 75E05 from S. frugiperda [id: 681368.1] flanking a 40 nts fragment of the ITR (2–40) and the ellipse represents 70 nts of the same BAC intercalated between a JcDV VP fragment (3,829–3,531) and the NS-ITR boundary (771–551). (C) BLAST alignment against lpNS of raw sequences obtained after amplifications of viral junctions from lpNS-transfected cells.

Briefly, the isolation of sequences containing mainly rearranged viral sequences is the signature of a dense recombination activity of cellular origin.

We repeatedly found, and verified, roughly the same viral rearrangement in each of the analyzed lpFull subclones (Fig. 6A) bringing together a NS-1/NS-2 fragment and a VP4 fragment, otherwise distant of ∼700 nucleotides in the originally transfected molecule, and placed in opposite orientations. Noteworthy, the considered VP4 fragment lies in the 3′ VP coding sequence which was displaced from the VP gene by the gfp insertion. Both went close to the p9 ITR. We also found molecules joining the same NS sequence with the same ITR nucleotides, without VP sequences. The integration situation thus seems to be rather simple in lpFull clones since only one rearrangement, was characterized from the different clones tested which are unlikely to be issued from the same transfected cell, even if we cannot exclude this possibility. Two non-viral sequences were identified interspersed with viral sequences (see below).

Five different lpVP clones and sub-clones were analyzed by walk-PCR and/or by reverse PCR (Fig. 6B). Two different sequences, shared by all the lpVP subclones and two clone-specific rearrangements were analyzed. All these sequences show the involvement of ITRs sequence fragments rearranged with NS and/or VP sequences fragments issued undoubtedly from the transfected linear vector. Among the few non-viral DNA sequences interspersed within the rearrangements, two were large enough to deserve identification (see below).

lpNS integrants are diverse in number, size and composition (Fig. 6C). They involve either ITR or ITR/NS sequences. Non-viral sequences, too short to be identified, alternate with vector-derived sequences, viral or not (gfp, pBR322, other).

Non-viral sequences are interspersed with viral ones in the rearrangements

The recent publication of two Spodoptera frugiperda genomes (Gouin et al., 2017) enriched the data base with pertinent sequences. All the non-viral sequences were compared to each other and to online databases NCBI (http://www.ncbi.nlm.nih.gov/BLAST/) (Altschul et al., 1997), SPODOBASE (http://bioweb.ensam.inra.fr/spodobase/) and Lepido DB (http://bipaa.genouest.org/blast/sfru/#) with an expected value > 10e5.

Figs. 6A–6C schematizes the location of cleaned viral sequences along the sequences of the transfected linear molecules, lpFull, lpVP and lpNS, respectively. No non-viral sequence (or not present in the transfected molecules) was identified in the lpNS rearrangements. Non-viral and unidentified sequences were found in the lpVP and lpFull rearrangements.

The sequence identified as “lpVP_1” as well as the sequence “lpVP_1/1” in Supplemental Information matched with numerous genomic sequences, best matches being with “Spodoptera frugiperda 41I04_SfBAC” [id: FP3404412.1] and “Spodoptera frugiperda 75E05_SfBAC” [id: FO681368.1].

The sequences identified as “lpFull_AP2/6a,” “lpFull_AP2/1a” and “lpFull_12/1” in the Supplemental Information match mainly with the “Spodoptera frugiperda sequence from bacterial artificial chromosome (BAC) clone 68E14” (GenBank: FO681381.1). Since these viral and cellular sequences are heavily modified and interspersed with unidentified small DNA sequences, we preferred to schematize their location in Fig. 6.

Last, the sequence identified as “lpFull_AP2/16a” in the Supplemental Information matches mainly with the “Spodoptera frugiperda” cell-line Sf9 ribosomal genes its1 and its2 (GenBank: Q478352.1). Note that in this case 63 nts of genomic sequence (1,326–1,389) overlap with 26 nts of viral sequences from the ITRs (2–28, 5,904–5,930), we cannot claim that it is a true junction.

Sequences “lpFull_AP2/7” and “lpFull_AP2/6b” exhibit also genomic sequences but the viral moiety of these sequences is too tiny to deserve mention. The BACs have been extensively described (d’Alencon et al., 2004, 2010), and, according to their organization, the sequences evoked before could pertain to highly repetitive, non-coding and intergenic regions. Using a preliminary version of TE annotation of the “Sf corn strain genome assembly” (Gouin et al., 2017), we found that, indeed, all the characterized flanking sequences are located in intergenic regions and/or different TE sequences which remain to be identified. Incidentally, the previously described flanking sequences of the circular vectors (Bossin et al., 2003) were also found located in intergenic and TE regions. However Sf genomic and transcriptomic data are currently insufficient to sustain the hypothesis of genomic rearrangements induced by JcDV integration vs. genomic sequences with identities to viral sequences as already evoked (Liu et al., 2011), whatever the cause.

NS proteins do not transactivate the p9 promoter

Considering only the results obtained with lpFull and lpVP clones, we seem to corroborate previous conclusions, i.e., in the presence of NS genes, integration of JcDV-derived sequences integrate within the host-cell genome at a low, probably single copy number although deletion of NS genes allows multiple copies integration, even if we cannot provide evidence of concatemers generation and integration from linear pVP molecules. However, as evoked in the same work and as reported for non-structural proteins from Parvovirinae (Batchu et al., 2001; Daeffler et al., 2003), a maintained expression of JcDV NS proteins may be harmful for the host cell and interfere with the cell cycle. Thus, it is not surprising to recover only cell clones having lost the capacity of NS expression after JcDV-based sequences integration. This is confirmed by the extensive rearrangements of NS coding sequences which rendered their expression unlikely. Moreover, a retrospective PCR amplification indicates that NS gene was already undetectable six days p.t. in lpFull-transfected cells (not shown). Similarly, cells transfected with pNS and lpNs have allowed the isolation of only some GFP+ clones (Fig. 2). However, in the DNA extracted from lpNS cell clones, numerous rearranged viral sequences interspersed with DNA fragments from a different origin, are accompanied with a sustained expression of NS mRNA. This could be explained by an expression of the NS genes low enough not to perturb the cell cycle. Actually, the XcmI restriction used to generate lpNS cuts the viral sequence between the p93 TATA box and the regulatory sequences (Dumas et al., 1992) and this could account a reduced expression of p93 driven mRNA. Our results are concordant with recently published data showing that numerous genomes are interspersed with parvovirus-related sequences (Liu et al., 2011) or endogenous viral elements (Katzourakis & Gifford, 2010; Theze et al., 2014) often fragmented/rearranged.

Nevertheless, we were unable to explain two results: (i) the percentage of GFP expressing cells increases with time in all the clones (Fig. 2), (ii) lpNS transformed cells seem to express GFP less stably and less efficiently than lpVP and lpFull transformed ones (Fig. 3B). We thus explored in vitro the regulatory activities of NS proteins on the p9 promoter.

Three plasmid vectors were built to express NS-1, NS-2 and NS-3, respectively. A reporter plasmid was adapted to place the expression of Luc under the control of p9 promoter and IPLB-Ld652 permissive cells were co-transfected to perform a “luciferase” assay matched against the Renilla luciferase expression. Fig. 7 clearly indicates that NS-1 expression does not significantly affect p9 activity even if we cannot ascertain that a functional amount of NS1 is produced. This is not completely unexpected since JcDV NS-1 seems to lack the COOH-terminus transactivation-associated region (Legendre & Rommelaere, 1994; Nuesch, 2006; Yang et al., 2006). Transactivation of the p9 promoter is neither devoted to NS-2 or NS-3 which on the contrary down-regulate it at 96 h. In addition, it is clear that the “basal” activity of the p9 promoter augments along with time, in vitro also. So, it is no more surprising that integration of linear molecules expressing GFP under the control of the p9 promoter is accompanied with a growing number of “green” cells, probably due to the growing number of cells which become detectable along with time. Also, the limited number of GFP expressing cells, even 30 days post transfection of lpNS linear vector could be explained by the retained capacity of these cells to express the NS genes.

Figure 7 JcDV NS proteins do not transactivate the p9 promoter in vitro.

IPLB-Ld652 permissive cells were co-transfected with plasmid vectors expressing JcDV NS-1, NS-2 and NS-3, respectively and a reporter plasmid adapted to place the expression of the firefly luciferase gene under the control of p93 promoter. The assay was matched against the renilla luciferase stable expression. Firefly luciferase activity was measured as indicated in the “Material and Methods” section. Three independent experiments were performed and the significance of the results was assessed after a Student’s test (R suite; R Core Team, 2014).

In vivo, injection of JcDV-derived plasmids in insect syncytial embryo gives rise to a somatic transformation of some cells throughout the larval stages (Royer et al., 2001). However, the modification does not cross the barrier of the germ cells. It remains to go further on the way of insect transformation, using linear JcDV-derived molecules equipped with a right ITR to integrate efficiently, a strong promoter, JcDV p9 being a good candidate, and an insect-compatible polyadenylation signal. Non-structural protein(s) expression, although probably useful for original single integration profile, are probably involved also in the rearrangement of both viral and cellular genomes. The Dependoparvovirus adeno-associated virus type 2 (AAV2) genome is site specifically integrated after interaction of Rep (NS) proteins and cellular proteins mainly involved in DNA repair (Nash et al., 2009), including nonhomologous end joining proteins (Daya, Cortez & Berns, 2009). Although the JcDV NS1 protein, as AAV2 Rep protein, recognizes and binds to a specific sequence of viral DNA (Ding et al., 2002), it does not seem capable of directing site-specifically the integration of the viral genome within the host genome. Cellular proteins could therefore be the major players in this integration. The fact that the flanking sequences are identified as “noncoding” may reflect spontaneous selection of cell clones least affected by the integration.

Conclusion

We investigated the status of JcDV-based linear DNA molecules after transfection in non-permissive Sf9 cells. Molecules harboring either complete JcDV sequences or ITRs and vp, i.e., the linear equivalent to the plasmids reported in (Bossin et al., 2003) or only p9-ITR and ns, the gfp being directly under the control of the p9 promoter, lead to GFP stable expression in transfected cells. However, JcDV sequence integration was systematically associated with major rearrangements (scrambling) even in the case of a possibly single integration event after transfection with complete JcDV-based molecule. Some genomic host-sequences were caught in the rearrangements and remain mostly unidentified even if some correspond to non-coding and repeated or transposable sequences. The lower number of cells expressing GFP in cells harboring a possible continuous expression of NS proteins as well as the obvious increase along with time of the number of GFP+ cells were intriguing. A luciferase assay of the NS-mediated regulation of the p9 promoter activity revealed that on the contrary to other Parvoviridae, NS-1 protein from JcDV seems not to transactivate the promoter driving the expression of structural proteins. In addition, NS-2 and NS-3 proteins down-regulate the p9 promoter late after transfection.

Altogether, it is possible to use JcDV-derived linear sequences, provided they harbor at least one ITR and poly-A signals, to direct the random integration of a reporter gene. The presence of ns genes could be facultative since integration seems mainly due to cellular factors.

Supplemental Information

Supplemental Information 1 Raw DNA sequences obtained after sequencing of the PCR amplificates.

Click here for additional data file.

The authors are indebted to M. Bergoin for the kind gift of original JcDV-based plasmids. J. Vlak is acknowledged for the gift of the Renilla luciferase vector and M. Ravallec for skillful assistance with microscopy. D. Mieulet is gratefully acknowledged for help with walk-PCR technique. Micheline Durand is acknowledged for her assistance during this study. Françoise-Xavière Jousset and Philippe Fournier are acknowledged for their guidance and useful critics. We also want to thank the referees for their helpful comments and suggestions.

Additional Information and Declarations

Competing Interests

Author Contributions

Data Availability

Sylvain Laverdure is a government employee in NIH working as a scientist at the Laboratory of Human Retrovirology and Immunoinformatics subcontracted by Leidos Inc.

Francine Rizk performed the experiments, analyzed the data, prepared figures and/or tables, authored or reviewed drafts of the paper, approved the final draft.

Sylvain Laverdure contributed reagents/materials/analysis tools, authored or reviewed drafts of the paper, approved the final draft.

Emmanuelle d’Alençon analyzed the data, authored or reviewed drafts of the paper, approved the final draft.

Hervé Bossin contributed reagents/materials/analysis tools, authored or reviewed drafts of the paper, approved the final draft.

Thierry Dupressoir conceived and designed the experiments, analyzed the data, prepared figures and/or tables, authored or reviewed drafts of the paper, approved the final draft.

The following information was supplied regarding data availability:

The raw data is supplied as a Supplemental File.

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
