# Peer review of "Linear Lepidopteran ambidensovirus 1 sequences drive random integration of a reporter gene in transfected Spodoptera frugiperda cells"

_PeerJ, doi:10.7717/peerj.4860_

## Round 0.1 · original submission · Minor Revisions

Please carefully follow the reviewers' proposals

Reviewer 1 ·

Basic reporting

No comment except that Figs. 4E-4G should be improved as indicated below in the 'General comments for the author'.

Experimental design

No comment.

Validity of the findings

No comment.

Additional comments

The paper describes molecular events occurred upon transfection of Sf9 cells with JcDV-based linear DNA.
The presented data are rather interesting and useful for future studies of related viruses. However, a minor revision of the presented paper is required before it could be accepted for publication.

Main points:

1. It is not clear what linear DNA molecules derived from pFull, pVP and pNS were used in transfection experiments. No information on this point can be found in Results, Materials and Methods, or Fig. 1. These linear DNA should be described unambiguously.

2. Figures 5E-5G are barely understandable. Another form for presentation of data on rearranged viral sequences should be found. For example, figures 5E-5G would be converted to a separate figure of a bigger size. Such a figure should clearly depict not only the positions of viral sequence fragments relative to the original sequence, but also show how these fragments are interconnected. Positions of non-viral sequences in these clones should be shown. In general, I would suggest avoiding output screens of sequence comparison software.

Minor comments:

Line 43 – Correct the word ‘analysed’.
Line 77 – “0,547 kb” looks strange, would be changed to 547 nucleotides.
Line 111 – What does the word “precise” mean in this context?
Lines 223-227 - should be removed
Line 239 – undetecTable

Reviewer 2 ·

Basic reporting

The manuscript is well written , conveys well the message.
The objectives are clear as well as the results .
The manuscript is written concisely, without undue details.

However, in the results section, it would be preferred not to repeat some parts of the materials and methods but to describe the outcome, for example:
When X construct cut with restriction enzymes Y and Z ...(the results were )....
replace with:
when X construct was cut with restictions enzymes to remove this (part of the genome) .... (the results were) proving that ...

Experimental design

Original research, clearly demonstrates that linear viral sequences may initiate integration in host genome.
The researchers had to use several constructs and trials in order to prove their hypotheses
Methods used are scientifically based

Validity of the findings

Data seems robust and conclusions well stated

Additional comments

This research is original, written in a well structure manner and deserves publication
However it may be better understood if the results are rewritten in a clearer manner as suggested above.

Annotated reviews are not available for download in order to protect the identity of reviewers who chose to remain anonymous.

---

## Round 0.2 · accepted · Accept

Thank you for completing the revision of the manuscript.

# Reviewer 1 ·

Basic reporting

No comment

Experimental design

No comment

Validity of the findings

No comment

Additional comments

With all the corrections made, the paper can be accepted for publication in PeerJ.